# PeerJ

# Effects of five southern California macroalgal diets on consumption, growth, and gonad weight, in the purple sea urchin *Strongylocentrotus purpuratus*

Matthew C. Foster[1], Jarrett E.K. Byrnes[1,2] and Daniel C. Reed[1]

[1] Marine Science Institute, University of California Santa Barbara, Santa Barbara, CA, USA
[2] Department of Biology, University of Massachusetts Boston, Boston, MA, USA

Corresponding author
Matthew C. Foster,
mcf543@gmail.com

## ABSTRACT

Consumer growth and reproductive capacity are direct functions of diet. Strongylocentrotid sea urchins, the dominant herbivores in California kelp forests, strongly prefer giant kelp (*Macrocystis pyrifera*), but are highly catholic in their ability to consume other species. The biomass of *Macrocystis* fluctuates greatly in space and time, and the extent to which urchins can use alternate species of algae or a mixed diet of multiple algal species to maintain fitness when giant kelp is unavailable is unknown. We experimentally examined the effects of single and mixed species diets on consumption, growth and gonad weight in the purple sea urchin *Strongylocentrotus purpuratus*. Urchins were fed single species diets consisting of one of four common species of macroalgae (the kelps *Macrocystis pyrifera* and *Pterygophora californica*, and the red algae *Chondracanthus corymbiferus* and *Rhodymenia californica* (hereafter referred to by genus)) or a mixed diet containing all four species *ad libitum* over a 13-week period in a controlled laboratory setting. Urchins fed *Chondracanthus*, *Macrocystis* and a mixed diet showed the highest growth (in terms of test diameter, wet weight and jaw length) and gonad weight, while urchins fed *Pterygophora* and *Rhodymenia* showed the lowest. Urchins consumed their preferred food, *Macrocystis*, at the highest rate when offered a mixture, but consumed *Chondracanthus* or *Macrocystis* at similar rates when the two algae were offered alone. The differences in urchin feeding behavior and growth observed between these diet types suggest the relative availability of the algae tested here could affect urchin populations and their interactions with the algal assemblage. The fact that the performance of urchins fed *Chondracanthus* was similar or higher than those fed the preferred *Macrocystis* suggests that the availability of the former could could sustain growth and reproduction of purple sea urchins during times of low *Macrocystis* abundance as is common following large wave events.

## INTRODUCTION

Sea urchins are dominant grazers in many benthic marine systems around the world and can exert a strong top-down influence on community structure (*Lawrence, 1975*). In kelp forests along the west coast of North America, strongylocentrotid sea urchins can have a large effect on the standing biomass of the giant kelp *Macrocystis pyrifera* and understory algal species (*North & Pearse, 1970*; *Leighton, 1971*; *Dayton, 1985*). The standing biomass of giant kelp, a preferred food of sea urchins (*Leighton, 1971*), fluctuates greatly in response to a range of physical and biological processes (*Dayton et al., 1999*; *Reed, Rassweiler & Arkema, 2008*; *Reed et al., 2011*; *Cavanaugh et al., 2011*), and when its abundance is low sea urchins are known to shift their diet to consume the remaining algal assemblage (*Ebeling, Laur & Rowley, 1985*; *Harrold & Reed, 1985*). Increased knowledge of the effect of diet on sea urchin consumption, growth and reproduction should lead to a better understanding of when and where they can have strong ecosystem effects.

The feeding rates, food selectivity, growth and reproduction of a variety of species of sea urchins vary with changes in quantity and types of available foods. This variation is due to consumer food preferences and the digestibility, absorption efficiency and composition of available food (*Lawrence, 1975*). Feeding and nutrient allocation to somatic and gonadal growth and gametogenesis also vary with diet, time of year, and environmental conditions (*Lawrence & Lane, 1982*). Some species of sea urchins exhibit strong food preferences in the presence of a mixture of algae (*Leighton, 1968*). Many perform better when consuming mixed diets (*Beddingfield & McClintock, 1998*; *Fernandez & Boudouresque, 1998*; *Fernandez & Pergent, 1998*; *Vadas et al., 2000*), suggesting that algal assemblage diversity could affect sea urchin performance. A comprehensive understanding of the interactions between sea urchins and macroalgal assemblages in any system requires knowledge of the factors that affect sea urchin feeding behavior and performance.

Here we experimentally evaluated the effects of algal diet on consumption, growth, and gonad weight of the purple sea urchin *Strongylocentrotus pupuratus* using four co-occurring species of macroalgae known to be part of its diet: the kelps *Macrocystis pyrifera* and *Pterygophora californica*, and the red algae *Chondracanthus corymbiferus* and *Rhodymeniacalifornica* (all algal species hereafter referred to by genus). These four species were chosen because they represented a large proportion (>75%) of the algal biomass in our study region in southern California (*Miller, Harrer & Reed, 2012*) and are consumed by purple sea urchins in the field (*Byrnes, Cardinale & Reed, 2013*). Thus, changes in the performance of purple sea urchins resulting from changes in the availability of these four species of macroalgae could have large implications for the structure of subtidal reef communities.

## METHODS

We measured algal consumption, test growth, jaw growth, change in whole body wet weight and gonad weight of urchins fed one of five experimental diets over an 89-day period in a controlled laboratory setting. Urchins used in the experiment were collected in October 2010 from a shallow (~4 m depth) boulder reef (34°24.9N 119°49.8W) located

offshore of the University of California, Santa Barbara. To minimize inherent variation in growth potential, consumption potential, and initial gonad weight, urchins were chosen to be relatively uniform in size (horizontal test diameter 33.5 ± 0.4 mm, mean ± SE) and presumably age (*Ebert, 1968*; *Ebert, 1977*; *Kenner, 1992*; *Russell, 1987*), and collected from a denuded urchin barren where their gonad weight was predicted to be uniformly low.

Upon collection, urchins were transported to the laboratory in insulated containers, placed in aquaria with running seawater and starved for one week prior to the start of the experiment. Blotted dry urchins (placed with the aboral end facing down on paper towels for 5 min) were weighed to the nearest 0.01 g and their horizontal test diameter was measured to the nearest 0.1 mm using Vernier calipers. To measure jaw growth, each urchin was injected with the fluorescent marker tetracycline following *Ebert (1982)*. 1.0 g of tetracycline was mixed with 100 mL of seawater, and 0.2 mL of the resulting solution was injected into each urchin through the peristomal membrane with a hypodermic needle. Tetracycline binds to actively calcifying tissues, effectively labeling jaw material present at the start of the experiment. Jaw material calcified after tetracycline administration was therefore unlabeled.

Each urchin was assigned to one of 35 labeled plastic containers (32 × 19× 11 cm) supplied with flow-through seawater. This setup allowed us to keep track of individuals without the use of external tags. Each container was supplied from the same head water tank; however, the containers were not connected and therefore water supplies were independent. Seawater temperatures ranged from 11.6 to 16.3 °C during the experiment and matched ambient conditions. Urchins were fed one of five macroalgal diets: a monospecific diet of either *Macrocystis pyrifera*, *Pterygophora californica*, *Rhodymenia californica*, or *Chondracanthus corymbiferus*, or an equal mixture of all four species (hereafter referred to as a mixed diet) with $n = 7$ urchins per diet type. Sea urchin containers assigned to each treatment were spread out randomly in space. Algae were added to the tanks during nine periods ranging in length from 4 to 8 days in which all experimental urchins were given a known amount of algae (on average either 34 g of one species, or in the mixed diet treatment 10 g of each of the four species). Sea urchins were given more algae than they consumed in all cases, except during one feeding period in which urchins in mixed diet treatments consumed all *Macrocystis* approximately 48 h before algae was removed and the feeding period terminated. During the 89-day experiment, urchins were exposed to algae for approximately 54 days. *Rhodymenia* was absent from the monospecific *Rhodymenia* treatment and from the mixed diet treatment for one of the feeding periods (14% of the total exposure time) due to its lack of availability in the field. Feeding periods were kept relatively short to prevent degradation of the algae, and algae were not immediately replaced due to logistical constraints on field collections. To study algal consumption, algal wet weight (after removing excess water with a spinning colander) was measured at the beginning and end of each feeding period. Consumption was calculated as wet weight (g) of algae consumed per urchin per hour using the total amount of hours urchins were exposed to algae (exposure time). We used consumption rate rather than amount consumed to standardize for different exposure times in the

*Rhodymenia* treatment. To evaluate the nutritional content of the algae, tissue samples of each species of algae were collected at three time points during the experiment and analyzed for carbon and nitrogen content (% dry weight). Samples were weighed wet (after removing excess water with a spinning colander), placed in a drying oven at 60 °C until dry, ground to a fine powder, and stored in a desiccator until analyzed by the UCSB Marine Science Institute Analytical Laboratory using the Dumas combustion method (duplicate samples from each species at each time point were tested). Mean carbon:nitrogen (C:N) ratios were calculated for each algal species and linear models were fit for each measure of urchin performance (see below) using C:N as the independent variable in each case.

At the end of the experiment the horizontal test diameter and wet weight (measured to the nearest 0.01 g in blotted dry urchins) of each urchin was measured. The change in test diameter and change in wet weight of each individual over the experiment was calculated by subtracting the initial value measured at the beginning of the experiment from that measured at the end of the experiment. Gonads were removed from each urchin upon dissection, placed in an oven at 60 °C until dry and weighed to the nearest 0.01 g. Final gonad dry weight was used as a measure of gonad growth because initial gonad weight was presumed to be nil as all individuals used in the experiment were similar in size and collected from a barren. We verified this assumption by taking eight urchins (test diameter $34.3 \pm 1.0$ mm (mean $\pm$ SE)) from the original collection site in the middle of the experiment (inadvertently, no data from the source population were taken at the beginning of the experiment) and measuring the mass of their gonads (see Results) following the same procedure described above.

Jaw growth was measured using half-pyramids of the aristotle's lantern following *Ebert (1982)*. Half pyramids were removed from each urchin and soaked in a 5% sodium hypochlorite solution for 24 h. For one half-pyramid per urchin, the total length from the oral tip to the flat shoulder at the aboral end (see *Ebert (1980a)* for pictures of points of measurement) was measured to the nearest 0.01 mm using a dissecting microscope equipped with an ultraviolet lamp and an ocular micrometer. Fluorescence from labeled tetracycline was observed from the oral tip to part way up the length of the jaw, indicating that this material had been present at the start of the experiment. Jaw growth was measured as the length of the non-fluorescent "band" extending from the top of the fluorescent area to the flat shoulder at the aboral end.

Differences among treatments were analyzed separately for each response variable (consumption rate (g of algae consumed $\cdot$ h$^{-1}$ averaged over the experiment), change in test diameter, change in jaw length, change in whole body wet weight, and final gonad dry weight) using one-way ANOVA. Diet selectivity was studied by examining the rate at which individual species of algae were consumed in mixture treatments by fitting a linear model with algal species as a fixed effect and container (individual urchin enclosure) included as a random effect (*Gelman & Hill, 2006*) as consumption rates of individual species of algae in a single container were not independent. After fitting linear models, assumptions of normality were tested by performing the Shapiro–Wilk test on the linear regression residuals, and homogeneity of variance was tested using Bartlett's test on the residuals

**Table 1 Performance and consumption across diets.** *F*-tables for linear models fit with different aspects of urchin performance as response variables and diet as a fixed factor.

| | | df | SS | MS | F | Pr(>F) |
|---|---|---|---|---|---|---|
| Change in test diameter | Diet | 4 | 56.2 | 14.1 | 7.52 | <0.001 |
| | Residual error | 30 | 56.0 | 1.87 | | |
| Change in wet weight | Diet | 4 | 0.642 | 0.160 | 11.1 | <0.001 |
| | Residual error | 30 | 0.434 | 0.014 | | |
| Gonad dry weight | Diet | 4 | 1.68 | 0.420 | 8.99 | <0.001 |
| | Residual error | 30 | 1.40 | 0.047 | | |
| Change in jaw length | Diet | 4 | 0.263 | 0.066 | 3.44 | 0.021 |
| | Residual error | 27 | 0.516 | 0.019 | | |
| Consumption rate | Diet | 4 | 0.559 | 0.140 | 19.1 | <0.001 |
| | Residual error | 30 | 0.220 | 0.007 | | |

across treatments (or across algal species, in the case of the mixed model). Homogeneity of varianace was not supported for the linear models of change in wet weight, consumption across treatments, and consumption within the mixed diet. ($p < 0.05$). Given that our data were from consumption and growth processes (implying multiplicative error) these three models were re-fit with log base 10 transformed data, and the resulting regression residuals exhibited normality and homogeneity of variance ($p > 0.05$). A post-hoc Tukey test was used to compare means (with statistical significance determined at the $p < 0.05$ level)—False Discovery Rate corrected *p*-values (*Benjamini & Hochberg, 2000*) were used in the case of the mixed model. For the log-transformed data, least-squares means and standard errors were back-transformed for plotting purposes. All statistical models were fit using R version 2.15-3 (*R Development Core Team, 2012*) with the nlme package for mixed models (*Pinheiro et al., 2012*) and the multcomp library for post-hoc analyses (*Hothorn, Bretz & Westfall, 2008*).

## RESULTS

Diet type had a significant effect on all performance measures (Table 1). Sea urchins fed *Chondracanthus, Macrocystis* and mixed diets exhibited the highest test growth, jaw growth, wet weight gain, and gonad weight, with no significant differences between these three diets (Fig. 1). Urchins fed *Pterygophora* exhibited significantly lower test growth compared to those fed *Chondracanthus* and *Macrocystis* diets, but had jaw growth and gonad weight that were not statistically different from either of them (Fig. 1). Urchins fed *Rhodymenia* exhibited the lowest values of all growth metrics and gonad weight (except that urchins fed *Pterygophora* had slightly lower mean test growth), with values significantly lower than those of urchins fed *Chondracanthus, Macrocystis* and mixed diets, in most cases (Fig. 1). Urchins collected from the field as a control group in the middle of the experiment had a gonad weight of $0.20 \pm 0.06$ g (mean $\pm$ SE) which was statistically not detectably different ($p = 0.24$, Welch two-sample *t*-test—the assumption

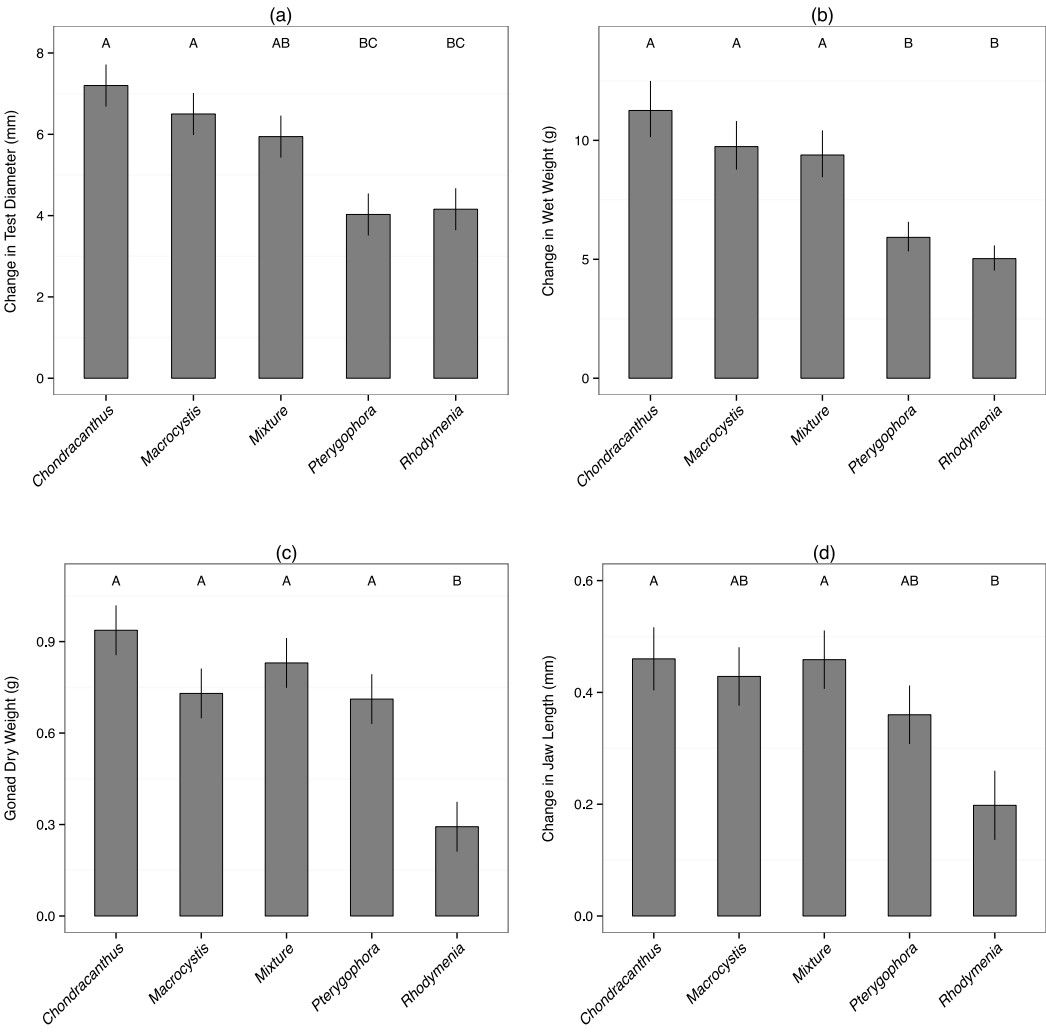

**Figure 1 Performance measures.** Bars represent mean (±1 SE) for (A) change in test diameter, (B) change in wet weight, (C) gonad dry weight, and (D) change in jaw length, over the course of the experiment. Letters indicate groups of means as determined by post-hoc general linear hypothesis tests with different letters signifying means that are different at the $p < 0.05$ level.

of normality was met for both samples (Shapiro–Wilk $p > 0.05$)—from the gonad weight of experimental urchins fed *Rhodymenia*, the least nutritious diet (Fig. 1C).

Within the mixed diet, sea urchins consumed *Macrocystis* at the highest rate, over twice as fast as any other species (Fig. 2A, Table 2). When algae were offered alone, however, the consumption rate of *Chondracanthus* and the mixed diet were similar to that of *Macrocystis* (Fig. 2B). In contrast, urchins consumed *Pterygophora* and *Rhodymenia* offered singly at the lowest rates.

The four species of algae used in the experiment differed in nutritional value as determined by their C:N ratios (Fig. 3). Importantly, we found no relationship between an alga's C:N ratio and food quality as measured by urchin performance ($p > 0.27$ for all correlations between the various measures of urchin performance and algal C:N

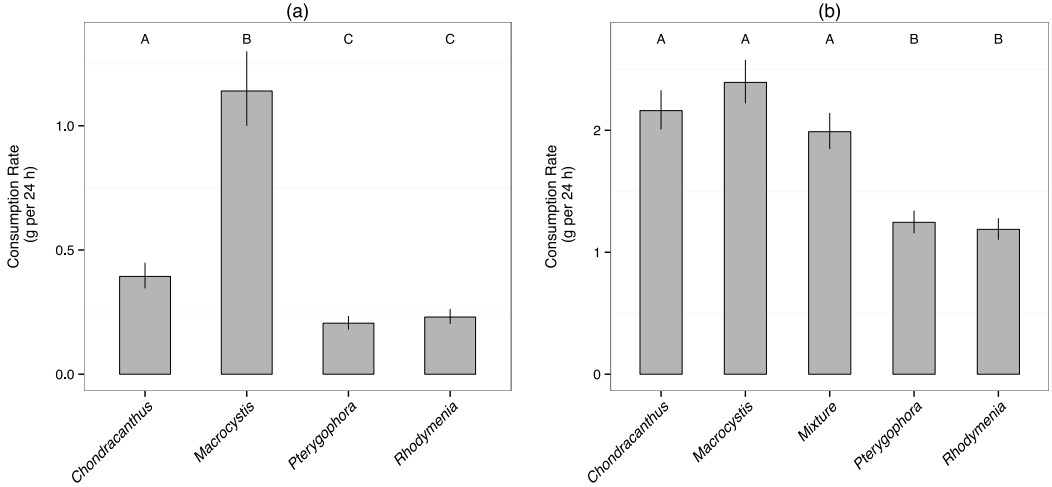

**Figure 2 Consumption rate.** Bars represent mean consumption rate (±1 SE) averaged over the experiment. Letters indicate groups of means as determined by post-hoc general linear hypothesis tests with different letters signifying means that are different at the $p < 0.05$ level. (A) Consumption rate of the different algal species in the mixed diet. (B) Consumption rate of all diets.

**Table 2 Consumption rate within the mixed diet.** *F*-table for a linear model fit with consumption rate as the response variable, algal species as a fixed factor and urchin container included as a random effect.

|  |  | df (numerator) | df (denominator) | F | Pr(>F) |
|---|---|---|---|---|---|
| Fixed effects | Species | 4 | 18 | 80.7 | <0.001 |
|  |  | **Standard deviation** |  |  |  |
| Random effects | Container | $1.05 \times 10^{-5}$ |  |  |  |
|  | Residual error | 0.150 |  |  |  |

ratio), with the exception of a correlation between gonad weight and algal C:N ratio (slope $= 0.04 \pm 0.02$ (estimate ± SE), $p = 0.01$). Counter to expectations, the species with the lowest C:N ratio (and thus the highest expected nutritional value) was *Rhodymenia*, which proved to be the least nutritional to urchins in terms of somatic and gonadal growth.

## DISCUSSION

We found test growth, wet weight gain, jaw growth, and gonad weight varied significantly among purple sea urchins as a function of diet. Overall, urchins fed monospecific diets of *Chondracanthus, Macrocystis*, and those fed a mixed diet grew significantly faster than those that were fed monospecific diets of *Pterygophora* or *Rhodymenia*, while urchins fed *Rhodymenia* had the lowest gonad weights. Urchins showed a strong preference for the naturally abundant *Macrocystis*, even when other algae were offered alongside. Our results suggest, however, that a diet consisting of other less preferred species of algae can sustain *S. purpuratus* at equally high levels of fitness at least over the short-term. This feature may be critically important in maintaining the reproductive capacity of purple sea

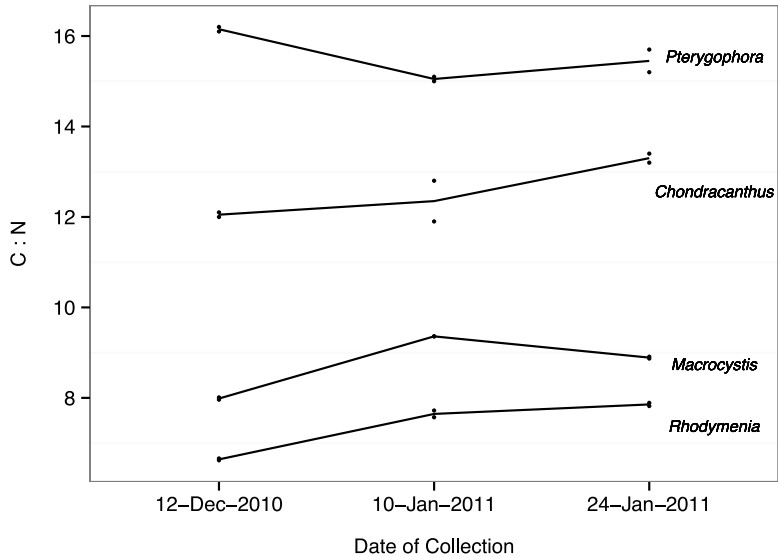

**Figure 3 Algal C:N.** Each path corresponds to a different a algal species (indicated on right), passing through the mean carbon:nitrogen ratio (C:N) calculated for each collection date. Individual points represent C:N values obtained for each species on each collection date ($n = 2$).

urchins during the peak winter spawning season when *Macrocystis* is least abundant due to intensive wave disturbance (*Reed, Rassweiler & Arkema, 2008*; *Reed et al., 2011*).

Since no differences in performance were observed among urchins fed *Chondracanthus*, *Macrocystis* (monospecific) and mixed diets, we found little evidence that purple sea urchins fed a mixed diet benefited over and above those fed a monospecific diet as long as either *Macrocystis* or *Chondracanthus* were present in the mixture. Because our particular mixed diet did not lead to increased performance, there appeared to be no benefit of diet complementarity (as assumed by the balanced diet hypothesis (*Pennings, Nadeau & Paul, 1993*)). Furthermore, since consumption was not higher in urchins fed a mixed diet, as could be permitted if species-specific toxins limited consumption of any one species, it does not seem likely that toxin minimization played a large role.

Despite these results, *Macrocystis* was consumed at the highest rate in mixed diets, suggesting that the effects of a mixed diet in the absence of this preferred food might prove different. Understanding urchin performance from algal mixtures in the absence of *Macrocystis* would be useful for understanding urchin dynamics after major kelp removal, as is common following large wave events. In this framework, *Byrnes, Cardinale & Reed (2013)* found that reduction in the abundance of sessile species by grazing purple sea urchins was positively correlated with species richness in plots where *Macrocystis* had been removed.

The low values for growth and gonad weight observed in purple sea urchins that were fed a monospecific diet of *Rhodymenia* occurred in spite of the fact that this species was often covered with epiphytic invertebrates such as hydroids and encrusting bryozoans, which have been shown to increase urchin somatic and gonad growth (*Knip & Scheibling, 2007* and references therein). The fact that this alga was absent from the *Rhodymenia* monospecific diet and mixed diet for 14% of the total exposure time (due to low

availability for collection) most likely did not bias results of performance measurements by much; test growth, jaw growth, change in wet weight, and gonad weight for urchins fed *Rhodymenia* were at least 40% lower than those for *Chondracanthus*, *Macrocystis* (monospecific), and mixed diets each case. Furthermore *Leighton (1968)* found test growth and gonad index were lower in purple sea urchins fed *Rhodymenia* than those fed either *Macrocystis* or *Pterygophora*. While our results with respect to urchins fed *Rhodymenia* should be interpreted cautiously, they suggest it is likely poor forage for purple sea urchins.

Like many organisms urchins display a trade-off in allocating resources to somatic vs. gonadal growth (*Lawrence & Lane, 1982*; *Steinberg & van Altena, 1992*). Our findings pertaining to the performance of urchins fed *Pteryoghora* are consistent with those of *Leighton (1968)*—purple sea urchins fed *Pterygophora* produced relatively large gonads and grew slowly during periods leading up to the peak spawning season. *Leighton (1968)*, however, showed evidence that somatic growth may increase as a proportion of total growth during the time period after spawning, highlighting that the effects of diet may vary with time of year among other factors. Additionally, concerning the jaw growth of urchins fed *Pterygophora*, *Ebert (1980a)* and *Levitan (1991)* related increased jaw size and decreased test size to lower food availability in two species of *Diadema*, suggesting resource allocation toward jaw growth facilitated food gathering ability. The fact that sea urchins in our experiment fed *Pterygophora* exhibited relatively high jaw growth relative to test growth provides preliminary evidence of a similar phenomenon in *S. purpuratus*. While the urchins in our experiment were not food limited, (as in *Ebert (1980a)* and *Levitan (1991)*), it seems possible that some effect of a *Pterygophora* diet could have triggered a similar mechanism as that by which food limitation caused resource allocation toward jaws in *Diadema*.

C:N ratio, used in our study as a rough measure of algal nitrogen and therefore protein content, was not correlated to any of the performance measures, as *Chondracanthus* and *Macrocystis*, which produced similar urchin performance had dissimilar C:N ratios (12.6 and 8.7, respectively, where the range of C:N values for all species was 7.4–15.6), suggesting that additional factors affected urchin performance. *Leighton (1968)* found that absorption efficiency was higher for purple sea urchins fed *Macrocystis* (70%) than *Pterygophora* (50%) or *Rhodymenia* (34%), and that protein and carbohydrates were absorbed more efficiently with *Macrocystis* than with *Pterygophora*. These results may explain some of the differences that we observed in performance as a function of diet. Concentrations of fatty acids, minerals (*Khotimchenko, Vaskovsky & Titlyanova, 2002*) and chemical deterrents (*Hall et al., 1973*; *Crews & Kho-Wiseman, 1977*; *Estes & Steinberg, 1988*; *Iken & Dubois, 2006*) also vary in other algae that are encountered by purple sea urchins and may play a role in nutritional quality, urchin food preference and consumption rates. Additional work is needed to uncover the relative contributions of different factors that may have led to the differences in consumption and performance that we observed.

We acknowledge that the sea urchins used in our experiment were collected from a single site, and the barren conditions present there suggest that the urchins had low food availability for some time prior to the experiment. Consumer history has been shown

to influence urchin somatic and gonad growth (*Livore & Connell , 2012*; *Williamson & Steinberg , 2012*),  and the proportion of time food is available has been shown to affect their grazing, growth and resource allocation (*Spirlet, Grosjean & Jangoux , 1998*). Thus, our results could have been affected by the prior history of the individual urchins used in our experiment. However *Leighton (1966)* found that purple urchins' food preferences did not vary among groups of individuals collected from a variety of habitats. This, coupled with the fact that *Leighton (1968)* observed the same rankings among *Macrocystis*, *Pterygophora*, and *Rhodymenia* in terms of promoting gonadal growth, and similar differences in allocations to somatic and gonadal growth in sea urchins fed *Pterygophora*, suggests that our results are generally applicable to purple sea urchins.

   *S. purpuratus* and the four algae we tested co-occur only along the west coast of North America (*Watanabe, 2010*). Nonetheless, strong preferences among algae, as well as marked effects of algal nutritional value in terms of fostering growth and reproduction in sea urchins have been noted in temperate and tropical systems around the world, including *Lytechinus variegatus* in the Gulf of Mexico (*Beddingfield & McClintock, 1998*), *Strongylocentrotus droebachiensis* in the Northwestern Atlantic (*Larson, Vadas & Keser , 1980*), *Centrostephanus rodgersii* and *Tripneustes gratilla* in Australia (*Steinberg & van Altena, 1992*) , *Paracentrotus lividus* in the Mediterranean and eastern Atlantic (*Boudouresque & Verlaque, 2007*), and various species of *Diadema* throughout the tropical Indo-Pacific (*Muthiga & McClanahan, 2007*). Such effects likely play a role in the overall consequences of grazing in many systems. For example, the food preferences of *Paracentrotus lividus*, which favor leafy algae over corallines or seagrasses have been shown to affect algal assemblage composition (*Boudouresque & Verlaque, 2007*). Likewise, *Scheibling & Anthony (2001)* suggested that *Strongylocentrotus droebachiensis'* preference for local brown algae over the invasive alga *Codium fragile* may spare patches largely composed of this alga that would otherwise be denuded under barren-forming conditions.

   Our results focus on the four species of algae that collectively comprised >75% of the biomass in kelp forests off Santa Barbara (*Miller, Harrer & Reed, 2012*) and elucidate the effects of these algae on performance of purple sea urchins, which may have implications for urchin populations in the wild. First, the ability to switch between diets, namely diets of *Chondracanthus* and *Macrocystis* (the urchin's preferred food), with little or no cost to growth and reproduction suggests that *Chondracanthus* could serve as an important alternative food source when *Macrocystis* is disproportionately removed by large waves (*Dayton & Tegner, 1984*; *Dayton et al., 1999*; *Gaylord, Denny & Koehl, 2008*). Upon the removal of *Macrocystis*, understory algae such as *Chondracanthus* become more abundant (*Arkema, Reed & Schroeter, 2009*; *Miller, Reed & Brzezinski, 2011*) and may serve as a suitable food that can sustain urchin populations and promote high growth and reproduction. *Chondracanthus'* relatively low rate of primary production (*Miller, Harrer & Reed, 2012*), however, indicates it might not be a long-term sustainable food source. Additionally our results suggest that in the context of the four abundant algae we tested, algal assemblage diversity may not be as important as the availability of one or two high quality food sources. We saw little evidence of a diversity effect; in mixed diets urchins

mostly consumed *Macrocystis*, and performed no differently than had they consumed *Macrocystis* alone.

Urchin preferences among these algae may also have implications for subtidal community structure, and more work is needed to better understand relative consumption rates of these dominant algae in nature (and the factors affecting these relative consumption rates, such as ambient oceanographic conditions and community interactions). Considering the model of urchin-algal dynamics presented by *Harrold & Reed (1985)*, where following the disappearance of *Macrocystis* urchins shift their behavior from occupying protected cracks and crevices while consuming drift *Macrocystis* to actively grazing the understory on the open substratum, the relative availability of the algae tested here could influence the extent of grazing that occurs after such shifts. Our experiment suggests that algal assemblage composition, along with total abundance and urchin density, all may play a role in shifting urchin dynamics in the wake of environmental perturbations to subtidal systems. More work is needed to understand whether these differences in diet translate to differences in urchin populations at the local and regional scale in pre- and post-disturbance temperate rocky reefs.

## ACKNOWLEDGEMENTS

Clint Nelson and Shannon Harrer assisted with field collections, and the efforts of many undergraduate workers were indispensable for data collection. Scott Simon and Christoph Pierre provided valuable advice on seawater flow-through setup. Robert Miller and Stephen Schroeter engaged us in many insightful discussions, and William Stockton offered advice on statistical analysis.

### Funding

Funding was provided by the National Science Foundation's Long Term Ecological Research Program and the University of California's Undergraduate Research and Creative Activities program. The funders had no role in study design, data collection and analysis, decision to publish, or preparation of the manuscript.

### Grant Disclosures

The following grant information was disclosed by the authors:
National Science Foundation's Long Term Ecological Research Program.
University of California's Undergraduate Research and Creative Activities program.

### Competing Interests

The authors declare there are no competing interests.

### Author Contributions

- Matthew C. Foster conceived and designed the experiments, performed the experiments, analyzed the data, contributed reagents/materials/analysis tools, wrote the paper, prepared figures and/or tables, reviewed drafts of the paper.

- Jarrett E.K. Byrnes conceived and designed the experiments, performed the experiments, analyzed the data, contributed reagents/materials/analysis tools, wrote the paper, reviewed drafts of the paper.
- Daniel C. Reed conceived and designed the experiments, contributed reagents/materials/analysis tools, wrote the paper, reviewed drafts of the paper.

### Field Study Permissions

The following information was supplied relating to field study approvals (i.e., approving body and any reference numbers):

Field collections were authorized through California Department of Fish and Game permit SC 11393.

### Data Deposition

The following information was supplied regarding the deposition of related data:

Long-Term Ecological Research (LTER) Network program.
10.6073/pasta/511fb37b21e90bb924b3ac691789c568.

### Supplemental Information

Supplemental information for this article can be found online at http://dx.doi.org/10.7717/peerj.719#supplemental-information.

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
