# Peer review of "Effects of five southern California macroalgal diets on consumption, growth, and gonad weight, in the purple sea urchin Strongylocentrotus purpuratus"

_PeerJ, doi:10.7717/peerj.719_

## Round 0.1 · original submission · Minor Revisions

Dear Matthew

This paper will be acceptable for PeerJ, but the helpful comments of both referees need to be addressed.

·

Basic reporting

no comments

Experimental design

no comments

Validity of the findings

no comments

Additional comments

The authors present the findings of an experiment on sea urchin consumption of different algal diets. Interactions between grazers and primary producers play a key role in shaping populations and communities in many marine ecosystems and understanding this process is critical. As such, the experiment is of interest to the marine ecology community, and the results shed some new light on dietary preferences and plasticity. The study is technically sound and well executed, and the results are interpreted appropriately and placed in a wider context. The paper is well written and the data presentation is good, allowing patterns to be easily deciphered. I have 2 main concerns and additional minor comments that the authors should consider, but overall I would recommend this paper for publication after a minor/moderate revision.

Main points:

1. The experiment, while well replicated at the individual level, was not replicated at the population level in that urchin collections were made just once at a single site. Given that the site is shallow, inshore and characterised by extensive barrens, I was left wondering about the generality of the findings, and whether feeding behaviour might differ in deeper, offshore populations that exist within more stable kelp forest habitat? Do the authors have any data (even preliminary) on urchin feeding rates/preferences from another site/population that might provide insights into the consistency of the observed pattern? If so, they should consider including it and, if not, perhaps they should include a caveat in the discussion about the generality of the results.

2. The paper is very America-centric, and I wonder if the authors might be able to broaden the discussion to increase the interest/readership for ecologists working in other systems. Is it possible to discuss the results in a wider context, by (i) comparing consumption rates of the purple sea urchin with those measured in other kelp dominated systems and (ii) comparing the degree of plasticity of the purple sea urchin with other urchins in different systems. I think there has been some work done on Centrostephanus, Tripneustes, Strongylocentrotus that might be comparable and provide some general insights into urchin food preference and/or consumption rates. If this is possible, a summary table would be very useful.

Minor points

Introduction, line 10: insert ‘many’ between ‘in’ and ‘benthic’ as the statement currently reads as if urchins are the dominant grazers in (all) benthic systems.
Line 63: were all containers fed seawater from the same system/header tank? Some more info on the experimental set up would be useful here....
Line 69: ‘Algae were added’ rather than ‘introduced’ reads better....
Line 109: ANOVA was used but how did the authors test for normality and homogeneity of variance? And how were the data treated if assumptions weren’t met?
Line 168: ‘Macrocystis’ typo
Line 223: ‘indicates’ not ‘indicate’

·

Basic reporting

The manuscript is well prepared, the writing style is accurate and unambiguous, is it pleasant to read. The topic is sufficiently well introduced and appropriately referenced. The discussion is restricted to the ecological perspective. Incidental relevance to the aquaculture of this species is worth a mention.

Experimental design

The manuscript describes the results of a 89 day experiment on 35 urchins fed one of 5 diets, so 7 individuals (replicates) per treatment. Firstly the authors must confirm these urchins had independent water supplies. Secondly, while the experiment seems small-scale, the use of fewer urchins is to be commended ethically and the fact individuals rather than populations are studied seems to have been successful, the error bars on the data suggest clear treatment effects.

Validity of the findings

The amount of new information is limited, from this relatively small experiment, but clearly presented and well supported by the data. There is a lack of time 0 data for gonad weight but the authors make a reasonable justification for their assumption that making a measure at half time gave a similar answer.

Additional comments

I have made some comments on the pdf attached

---

## Round 0.2 · accepted · Accept

Thank you for addressing the referees comments. This is fine for publication now.